# A CRISPR-Cas13b System Degrades SARS-CoV and SARS-CoV-2 RNA In Vitro

**DOI:** 10.3390/v16101539

**Published:** 2024-09-28

**Authors:** Klara Andersson, Ani Azatyan, Martin Ekenberg, Gözde Güçlüler, Laura Sardon Puig, Marjo Puumalainen, Theodor Pramer, Vanessa M. Monteil, Ali Mirazimi

**Affiliations:** 1Department of Laboratory Medicine, Unit of Clinical Microbiology, Karolinska Institutet, 17177 Stockholm, Sweden; andersson.klara.linnea@ki.se (K.A.); ali.mirazimi@ki.se (A.M.); 2Biomedrex Genetics, Alfred Nobels allé 8, 14152 Stockholm, Sweden; ani.azatyan@biomedrex.com (A.A.); martin.ekenberg@biomedrex.com (M.E.); gozde.gucluler@ki.se (G.G.); laura.sardon.puig@ki.se (L.S.P.); marjo.puumalainen@biomedrex.com (M.P.); theodor.pramer@biomedrex.com (T.P.); 3Public Health Agency of Sweden, 17182 Solna, Sweden; 4National Veterinary Institute, 75189 Uppsala, Sweden

**Keywords:** SARS-CoV, SARS-CoV-2, antiviral development, CRISPR-Cas13b

## Abstract

In a time of climate change, population growth, and globalization, the risk of viral spread has significantly increased. The 21st century has already witnessed outbreaks of Severe Acute Respiratory Syndrome virus (SARS-CoV), Severe Acute Respiratory Syndrome virus 2 (SARS-CoV-2), Ebola virus and Influenza virus, among others. Viruses rapidly adapt and evade human immune systems, complicating the development of effective antiviral countermeasures. Consequently, the need for novel antivirals resilient to viral mutations is urgent. In this study, we developed a CRISPR-Cas13b system to target SARS-CoV-2. Interestingly, this system was also efficient against SARS-CoV, demonstrating broad-spectrum potential. Our findings highlight CRISPR-Cas13b as a promising tool for antiviral therapeutics, underscoring its potential in RNA-virus-associated pandemic responses.

## 1. Introduction

Infectious diseases caused by viruses have had catastrophic consequences for our society, recently demonstrated by the COVID-19 pandemic, caused by the coronavirus, Severe Acute Respiratory Syndrome virus 2 (SARS-CoV-2). RNA viruses, like SARS-CoV-2, exhibit high mutation rates, allowing them to evolve within their host and to acquire high-level persistence, which leads to an immune response escape, high virulence, and a higher degree of cross-species transmission [1,2]. Consequently, RNA viruses are a threat to human health, as they can lead to the inefficiency of existing antivirals and vaccines.

Coronaviruses are a group of RNA viruses that are continually re-emerging. Before the outbreak of Severe Acute Respiratory Syndrome virus (SARS-CoV) in 2002, corona-viruses were known to cause mild upper respiratory tract infections [3]. SARS-CoV challenged this view by causing lower respiratory infections with severe complications. The SARS-CoV epidemic was followed in 2012 by another coronavirus epidemic caused by the Middle East Respiratory Syndrome virus (MERS-CoV), which, since then, has been responsible for sporadic, local outbreaks in the Middle East. In 2020, a new coronavirus emerged, SARS-CoV-2, which reached a pandemic level. Due to its zoonotic potential, respiratory transmission, and relatively high mutation rates, it is likely that epidemic and, in worst cases, pandemic events caused by coronaviruses will continue to appear in the future [3].

To date, there are no effective countermeasures against coronaviruses and even other emerging RNA viruses. The recent SARS-CoV-2 pandemic highlighted the importance of a rapid development of antiviral treatments. CRISPR-Cas13b is a suitable candidate to target viral RNA [4,5,6,7,8]. Cas13b, of CRISPR-Cas Type VI, is an RNA-targeting, RNA-guided effector capable of efficient RNA degradation in mammalian cells [9,10,11,12]. Cas13b is guided by CRISPR-associated RNA, referred to as guide RNA (gRNA), which can be designed to target a wide variety of sequences [10,13,14]. gRNAs consist of a spacer sequence, a 30-nucleotide-long sequence that is complementary to a target sequence, and a direct-repeat of 36 nucleotides that serves as a recognition site for Cas13b [15].

It has previously been shown that CRISPR-Cas13b can degrade SARS-CoV-2 [5,6,7,16]. Here, we present a platform to design 30-nucleotide-long gRNAs directed at RNA viruses, in a complex with CRISPR subtype Cas13b, which have proved their high target specificity and RNA knockdown in previous studies [5,17]. We can confirm the efficiency of our CRISPR-Cas13b-associated gRNAs against one variant of live SARS-CoV-2, SWE:01:2020 (Wuhan strain; GenBank MT093571). We also demonstrated the broad-spectrum potential of our system by targeting the genetically different SARS-CoV with the same system, which, to our knowledge, has not been achieved before.

## 2. Materials and Methods

### 2.1. Cell Culturing

Human embryonic kidney cells expressing angiotensin-converting enzyme 2 (HEK293/ACE2) (GenScript, Cat No M00770) were cultured in 10% fetal bovine serum (FBS) supplemented in DMEM (Life Technologies, Carlsbad, CA, USA). Cells were routinely passaged, incubated at 37 °C, 5% CO_2_, and regularly tested against mycoplasma.

### 2.2. Viral Strains

Severe Acute Respiratory Syndrome-Related Coronavirus 2 (SARS-CoV-2) isolate SWE:01:2020 (GenBank: MT093571) and Severe Acute Respiratory Syndrome-Related Coronavirus (SARS-CoV) isolate Frankfurt:01:2003 (Genbank: AY291315.1) were handled in the bio safety level 3 lab of Folkhälsomyndigheten, Stockholm.

### 2.3. Biosafety

All experiments involving SARS-CoV and SARS-CoV-2 were performed in a Biosafety Level 3 laboratory in compliance with the Swedish Public Health Agency guidelines (Folkhälsomyndigheten, Stockholm).

### 2.4. SARS-CoV-2 Genomes Used for gRNA Design

Out of the 500,000 publicly available genomes at the time, 9 February 2022, from GenBank, 40,000 NGS SARS-CoV-2 genomes were used for gRNA design. This included all the variants circulating at the time.

### 2.5. gRNA Design

The 30-nucleotide-long gRNA sequences were selected based on a proprietary gRNA discovery platform developed by Biomedrex. This AI-based discovery platform consists of modules that evaluate potential targets in viral sequences based on specific criteria to select the most promising gRNA candidates for experimental testing. To protect the algorithms, no detailed publications or patents are currently provided. In essence, the selection criteria include parameters such as target site conservation on the viral genome, off-target overlap against the human genome, and the secondary structure of the gRNAs. The latter plays a crucial role in the efficient complexation of the gRNA and Cas protein and the subsequent binding of the CRISPR Cas complex with the corresponding target sequence.

In total, 7 genes on the SARS-CoV-2 genome were prioritized for targeting. All genes encoding structural proteins were chosen, including Membrane (M), Nucleocapsid (N), Envelope (E), and Spike (S). In addition, several genes encoding non-structural proteins were also considered: NSP3, NSP5, and NSP12.

Using the discovery platform, the 40.000 NGS SARS-CoV-2 genomes from GenBank were mined for gRNAs. The genomes were aligned, and the degree of conservation was calculated at each nucleotide position to obtain target genomic region conservation. The genomes were also aligned with the human genome to avoid human transcriptome overlap. Other parameters such as sequence specificity and the secondary structure of the gRNA sequence, as well as the chance and accuracy of direct repeat hairpin-loop formation for CRISPR Cas complex formation, were of decisive importance as well.

### 2.6. Cloning of Guide RNA Plasmids

The gRNA plasmid pC0043-pspCas13b backbone was from Addgene (Cat No 103854) [18]. The gRNA sequences were then Gibson-cloned into the PC0043 backbone, where the gRNA sequences were inserted as long reverse primers using the Gibson Assembly Cloning Kit (NEB, Cat No E5510S).

Fwd primer 5′-GTTGTGGAAGGTCCAGTTTTGAG-3′;

Rev primers 

M:3′-CTGGACCTTCCACAACCCTTTTAATTGAATTAATATATGTTTTTGGGGTGTTTCGTTCCTTTCCACA-5′NSP3 (1):3′-CTGGACCTTCCACAACCTGATAGTAGTAGATTGGTTAGAAGAAGAAGGTGTTTCGTTCCTTTCCACA-5′NSP3 (2):3′-CTGGACCTTCCACAACGTTGAATCCCAGTTAAAGACATGTTTGTTGGGTGTTTCGTTCCTTTCCACA-5′NSP3 (3):3′-CTGGACCTTCCACAACCCTCCCATTTTTCTTGTTATGTATACA CTTGGTGTTTCGTTCCTTTCCACA-5′NSP3 (4):3′-CTGGACCTTCCACAACTAGGAAAGTAGTTCAAGTTTTCACTATAAGGGTGTTTCGTTCCTTTCCACA-5′N:3′-CTGGACCTTCCACAACCCCTTAAATTCCAGAAGGAACG GTACAACTGGTGTTTCGTTCCTTTCCACA-5′NSP5 (1):3′-CTGGACCTTCCACAACCCATTAAGGTATACCACGTACATTGTTTTTGGTGTTTCGTTCCTTTCCACA-5′NSP5 (2):3′-CTGGACCTTCCACAACCTTGTAATGGTCGGACATGGTTCTTTAATAGGTGTTTCGTTCCTTTCCACA-5′NSP5 (3):3′-CTGGACCTTCCACAACTTTTGCCGTTAAGGTCAAACTCGTCTTTCTGGTGTTTCGTTCCTTTCCACA-5′S (1):3′-CTGGACCTTCCACAACCAGACTGAAGTAGTGGAGATTAATGTTTACGGTGTTTCGTTCCTTTCCACA-5′S (2):3′-CTGGACCTTCCACAACCCACCACAAAACATTTAAACAA ACTGAACAGGTGTTTCGTTCCTTTCCACA-5′S (3):3′-CTGGACCTTCCACAACTCCCATTATTTGTGGTGCACACTTTCTTAAGGTGTTTCGTTCCTTTCCACA-5′S (4):3′-CTGGACCTTCCACAACCCCATTATTTGTGGTGCACACTTTCTTAATGGTGTTTCGTTCCTTTCCACA-5′S (5):3′-CTGGACCTTCCACAACCCATTAAATATTAATATTAGTCGTTAGAAAGGTGTTTCGTTCCTTTCCACA-5′NSP12 (1):3′-CTGGACCTTCCACAACCCGTAATTGTTACTTATTATTCTTAGATGTGGTGTTTCGTTCCTTTCCACA-5′NSP12 (2):3′-CTGGACCTTCCACAACCTACTTTGACAGATAACCAGTATCATGATGGGTGTTTCGTTCCTTTCCACA-5′NSP12 (3):3′-CTGGACCTTCCACAACCCATTCCTTCCATGTGTATTAGTAGTGGGAGGTGTTTCGTTCCTTTCCACA-5′NSP12 (4):3′-CTGGACCTTCCACAACCCCTACTGTAATGCAAAACATATACGCTTTGGTGTTTCGTTCCTTTCCACA-5′NSP12 (5):3′-CTGGACCTTCCACAACCGCTCGTTCTTGTTCACTCCGGTATTAAGAGGTGTTTCGTTCCTTTCCACA-5′NSP12 (6):3′-CTGGACCTTCCACAACATTGCTATCATCAGTATTAGCGACTATCGTGGTGTTTCGTTCCTTTCCACA-5′

### 2.7. pspCas13b Plasmid

The pspCas13b plasmid, pC0046-EF1a-PspCas13b-NES-HIV, was a gift from Feng Zhang (Addgene plasmid, Cat No 103862) [18].

### 2.8. Production of gRNA and Cas13b Plasmids

A total of 10 ng of plasmids were transformed into 25 µL competent *E. coli* cells (OneShot, Cat No C404010) by heat shock for 30 s at 42 °C. The bacteria were then incubated in a 300 µL SOC medium for 1h at 37 °C in a shaking incubator of 225 rpm. Then, 50 µL of the mixture was spread on LB agar plates containing 100 µg/mL ampicillin and incubated overnight at 37 °C. Colonies were picked the day after and incubated in 5 mL LB broth media containing 100 µg/mL ampicillin for 4–8 h at 37 °C in a shaking incubator of 225 rpm. The mixture was then moved into 100 mL LB broth media containing 100 µg/mL ampicillin and incubated for 18 h at 37 °C in a shaking incubator of 225 rpm. Plasmid purification was performed using Qiagen’s HiSpeed plasmid Midi kit (Cat No 12643), according to Qiagen’s protocol. The plasmid sequences were confirmed with Sanger sequencing.

### 2.9. Reverse Transfections of Cas13b and Single gRNA Plasmids in HEK293/ACE2 Cells

For in vitro screenings, HEK293 cells expressing ACE2 (GenScript, Cat No M00770) were used. A total of 100 ng of Cas13b and 100 ng of gRNA plasmid were prepared in opti-MEM (ThermoFisher, Carlsbad, CA, USA Cat No 31985062) and mixed with GeneJammer transfection reagent (Agilent Cat No 204130), according to the manufacturer’s protocol. 20 µL of the mix was distributed per well of a Poly-D-lysine (Thermo Fisher, Cat No A3890401) coated 48 well plates (CytoOne, Cat No CC7682-7548). Then, 50,000 HEK293/ACE2 cells per 200 uL DMEM media, 10% FBS, were seeded on top of the DNA-GeneJammer mix and incubated at 37 °C, 5% CO_2_, for 48 h. For each experiment, a non-targeting gRNA (NT) that did not target SARS-CoV-2 was used. The silencing effect of each gRNA was normalized to the non-targeting control.

### 2.10. Combination of Guide RNAs in HEK293/ACE2 Cells

A total of 50 ng of each gRNA plasmid was used in the combination of two gRNAs. This was compared to 50 ng of each gRNA plasmid in the pool, delivered individually. The Cas13b was kept constant at 100 ng throughout the experiment. The NT control remained the same as in the single gRNA experiment—100 ng gRNA plasmid and 100 ng Cas13b plasmid. GeneJammer reverse transfections were also used for the combination experiment.

### 2.11. Infection Assays

The transfected cells in 48-well plates were infected 48 h post transfection. Cells were infected for 1h with SARS-CoV-2 MOI 0.1 in 2% FBS DMEM. Cells were then washed with PBS, supplemented with 5% FBS DMEM, and recovered 24 h post infection. The cells were recovered in TRIzol (Invitrogen, Waltham, MA, USA, Cat No 15596026) for qPCR analysis.

### 2.12. RNA Extraction and qPCR of Infected Samples

RNA was extracted from the TRIzol-inactivated samples using a direct-zol RNA Miniprep kit (Zymo research Cat No R2050). qRT-PCR was performed using TaqMan Fast (Applied biosystems, Cat No 4444556). qRT-PCR was performed with a StepOne real-time PCR system (Applied biosystems, Waltham, MA, USA, Cat No 4376600) under the following conditions: reverse transcription, 10 min at 50 °C; denaturation, 2 min at 95 °C; amplification 45 cycles of 10 s at 95 °C for the denaturation and 40 s at 60 °C for the annealing/extension step. For the analysis of SARS-CoV-2 targeting the E gene of SARS-CoV-2, the following primers and probes were used:

Fwd: 5′-ACAGGTACGTTAATAGTTAATAGCGT-3′;

Rev: 5′-ATATTGCAGCAGTACGCACACA-3′;

Probe: FAM-ACACTAGCCATCCTTACTGCGCTTCG-MGB.

For the analysis of SARS-CoV, the following primers and probes were used, targeting the E gene of SARS-CoV:

Fwd: 5′-TGCCTCTGCATTCTTTGGA-3′;

Rev: 5′-TAAGTCAGCCATGTTCCCG-3′;

Probe: FAM-CACGCATTGGCATGGAAGTCACA-TAMRA.

In both cases, RNAse P was used as an endogenous gene control, detected with the following primers and probes:

Fwd: 5′-AGATTTGGACCTGCGAGCG-3′;

Rev: 5′-GAGCGGCTGTCTCCACAAGT-3′;

Probe: FAM-TTCTGACCTGAAGGCTCTGCGCG-MGB.

### 2.13. Data Analysis

Three independent biological experiments with three technical replicates were used throughout this study. All data visualizations and analyses were carried out using GraphPad Prism software version 10. A *p*-value of *p* < 0.05 was considered statistically significant.

## 3. Results

### 3.1. Selection of Guide RNAs Using a Novel AI Tool

Coronaviruses contain a large genome of around 30,000 nucleotides, giving nearly as many potential target sites for the design of the 30-nucleotide-long gRNA sequences. Several studies showed various applications for designing gRNAs for CRISPR-Cas13b [15,19,20]. More importantly, some studies designed gRNAs to degrade viruses [21,22,23], and SARS-CoV-2 in particular [5,16,24]. Here, we developed a novel AI tool to design and select SARS-CoV-2-targeting gRNAs compatible with Cas13b.

From the available SARS-CoV-2 genomes, a subset of 40,000 genomes were analyzed. We selected seven genes to consider for targeting: all of the structural proteins, Membrane (M), Nucleocapsid (N), Envelope (E), and Spike (S), and three of the non-structural protein NSPs: NSP3, NSP5 and NSP12. After application of specific criteria, gRNA candidates were ranked, and the 20 top-ranked gRNAs were selected (Figure 1). The selected gRNA candidates targeted different sites of all the considered genes, except from E, where no gRNAs had a high enough ranking for selection. The selected gRNAs’ target gene and sequences are shown in Table 1.

### 3.2. Validation of the gRNAs

To validate the effect of the top-ranked gRNAs against SARS-CoV-2 in mammalian cells, we used the gRNAs together with the subtype pspCas13b, originated from *Prevotella* sp. P5–125 [18]. HEK293-ACE2 cells were transfected with two plasmids: one containing the gRNA sequence and one containing the cas13b sequence (100 ng of each). A non-targeting gRNA (NT) was used as a control.

48-h post transfection, the cells were infected with SARS-CoV-2 at a multiplicity of infection (MOI) of 0.1 and then recovered 24 h post infection (Figure 2a). The silencing effect of the gRNAs was defined by measuring the effect on the quantity of viral RNA using qRT-PCR. Out of the 20 gRNAs, 12 proved to have a silencing effect on SARS-CoV-2 (Figure 2b, Table 2). The gRNAs are summarized and ranked in Table 2; at best, a NSP12-targeting gRNA led to a 63% decrease in SARS-CoV-2 infection compared to the NT control (Figure 2b, Table 2).

### 3.3. gRNAs in Combination

After yielding a ranked list of the best gRNAs (Table 2), we wanted to test if combining gRNAs could potentiate the antiviral effect of gRNAs separately. Previous studies have shown that combinations of gRNAs increase RNA degradation while also providing a system that is more resilient to mutations [5,6].

The CRISPR-Cas13b complex is only active if the plasmids encoding Cas13b and gRNA are present in the same cell [7,17]. When using separate plasmids to express Cas13b and gRNA, there is a risk that the plasmids will not be located in the same cell. The complexity of the colocalization increases with the addition of more plasmids; thus, we restricted the combination to two different gRNA plasmids.

From the top-performing gRNAs, we selected gRNAs targeting different genes (NSP12, NSP5, S, and N) (Table 3). To maintain the final gRNA plasmid quantity in the pool, 50 ng of each gRNA plasmid was used to reach a final quantity of 100 ng, consistent with the amount in the previous single gRNA experiments.

The combination of NSP12 (2) and NSP5 (1) proved to significantly decrease viral RNA in cells when compared to the individually delivered gRNAs (Figure 3a). A combination of S (3) and NSP12 (2) resulted in a significant decrease when comparing to the single NSP12 (2), but not the single S (3) (Figure 3b). When combining NSP12 (2) with N, the combination outperformed the single N, but not the single NSP12 (2) (Figure 3c).

### 3.4. Efficiency against SARS-CoV

Next, we wanted to explore the broadness of our system by using the same gRNAs that were designed to target SARS-CoV-2 to its relative, SARS-CoV. Although SARS-CoV and SARS-CoV-2 exhibit phylogenetic relatedness, their percentage of sequence identity is around 80% [25,26]. We explored the mismatches between SARS-CoV and the four previously validated gRNAs from the combination experiment by aligning the gRNAs to SARS-CoV (Table 3). The best matching gRNA, NSP12 (2), only had one mismatch to the SARS-CoV NSP12 corresponding sequence, whereas N had four mismatches, NSP5 (1) had seven mismatches, and S (3) had twelve mismatches (Table 3). When tested for efficiency against SARS-CoV, NSP12 (2) showed a decrease on SARS-CoV RNA of 35%, whereas N exhibited a decrease of 22%. NSP5 (1) and S (3) had no effect (Figure 4, Table 4).

## 4. Discussion

Emerging viruses have the potential to rapidly spread and host escape by frequent mutations, which poses a great challenge in antiviral development. Most licensed antivirals are designed to target proteins encoded by the virus [27]. While this approach allows for a high degree of specificity in combating the virus, it also comes with the downside of increasing the likelihood that the virus will develop resistance to the antiviral treatment over time [27]. However, we also have access to broad-spectrum antivirals that target common viral properties found in several viruses. Nucleoside analogs are often considered to be broad-spectrum antivirals and commonly target viral enzymes or processes essential for viral replication [28,29]. In this study, we focused on bringing CRISPR-Cas technology as a new strategy against viral diseases and, particularly, emerging viral diseases.

CRISPR-Cas13b has the potential to work as a broad-spectrum strategy by targeting conserved viral genomic material [6]. In addition to this, CRISPR-Cas13b has a programmable feature that would allow a few modifications in the gRNA design to target a new, emerging RNA virus [30]. Previous studies show that CRISPR-Cas13b has a mismatch tolerance, which was confirmed by our study, and this implies that the system is resilient to frequently mutated viruses [5,7]. Given these considerations, CRISPR-Cas13b could potentially play an important role in combating emerging RNA viruses. In this study, we built a system based on 40,000 genomes of SARS-CoV-2 and tested it on one variant. It could be interesting to test several SARS-CoV-2 variants, which has been achieved in previous studies [5,7].

We showed that a two-plasmid system to deliver the CRISPR-Cas13b complex could be used for the silencing of viral RNA. However, this does come with some limitations. The more plasmids that are introduced, the harder it becomes to ensure and control their simultaneous presence in the cell. Anchoring a fluorophore to each plasmid could be a way to observe cell colocalization, but this would lead to a complex experimental design when combining numerous plasmids. It is not certain that a significant number of cells will simultaneously receive all the plasmids in such an experiment, making combinations of multiple plasmids inefficient. Even combining two plasmids was challenging, as we were unable to monitor colocalization and determine the number of cells that received both plasmids. Instead of using several plasmids, it could be beneficial to change the structure of the system. One way to ensure the simultaneous colocalization of Cas13b and gRNA could be to design a one-plasmid system that would encode both Cas13b and gRNA. Another possibility is to change the delivery method completely. Plasmid delivery worked well for screening gRNA candidates in cell lines. For a more complex model, like animal models, mRNA delivery has proven to be a safe and efficient delivery method of CRISPR-Cas13b [4,31,32].

Despite the complexity of the cell location when introducing multiple plasmids, we could show that combining two gRNAs could significantly enhance viral silencing, which is in line with previous studies [5,6,33]. A combination of the top-performing gRNAs, NSP12 (2) and NSP5 (1), outperformed their single counterparts, indicating that combining these gRNAs together results in a synergy that enhances viral silencing (Figure 4).

We used a combination of different guide RNAs, but creating a combination treatment with CRISPR-Cas13b and a drug with a different mechanism would also be possible. Using CRISPR-Cas13 systems together with other drugs has been achieved before, for example, with small-molecule drugs such as Remdesivir [6]. Another thing that could be used in combination with a CRISPR-Cas13 approach is an immunomodulating agent that would enhance the host’s antiviral response [34]. Individuals with severe COVID-19 have immune response delay and would therefore need immunomodulators that could boost the immune response, in combination with antivirals to treat the infection [34].

We showed that the same CRISPR-Cas13b system designed to target SARS-CoV-2 could also silence SARS-CoV and that the efficiency of the silencing was dependent on the number of mismatches. A mismatch of one or four nucleotides was tolerated, whereas mismatches of seven and twelve nucleotides abolished viral silencing (Figure 4, Table 4). A study about CRISPR-Cas13b as a system confirms that RNA knockdown is achievable when using pspCas13bs and gRNAs with up to four mismatches [35], which aligns with our findings.

NSP5 is a rather conserved viral protein, and we therefore expected fewer mismatches than the seven mismatches that were observed here. It might be expected that a gRNA targeting a non-structural protein like NSP5 would have fewer mismatches than a gRNA targeting a structural protein such as N, but N is quite conserved for being a structural protein [36,37]. What was expected was that the gRNA targeting S had the most mismatches, since this region in the SARS-CoV-2 is known to mutate quite frequently [37]. Some studies have used CRISPR-Cas13b to simultaneously target different SARS-CoV-2 strains and coronaviruses that cause the common cold, but, to our knowledge, our study is the first one using CRISPR-Cas13b to simultaneously target SARS-CoV-2 and SARS-CoV [5,6,7].

To achieve efficient viral silencing and a broad-spectrum effect, the choice of the target site is essential. Initially, all genes encoding structural proteins were selected as gRNA candidates since they are critical for virus infectivity and propagation [36,37]. Non-structural proteins are crucial for viral replication and are therefore interesting targets when designing antiviral approaches [37]. We also know that non-structural proteins generally are more conserved comparing to structural proteins that are more exposed for the host’s immune system [37]. From the non-structural proteins, NSP3, NSP5, and NSP12 were selected as gRNA candidates, since they have a high degree of conservation and are crucial for viral replication [38,39,40]. We were most successful with targeting the gene encoding NSP12, which proved to be rather conserved between SARS-CoV-2 and SARS-CoV, and had the fewest mismatches to SARS-CoV, which could explain its great effect on SARS-CoV. NSP12 is crucial for viral replication since it constitutes the important RNA-dependent RNA polymerase (RdRP) and has proven to be a promising target in previous studies, which is also why it was initially considered [41,42,43]. In comparison to NSP3 and NSP5, NSP12 has a direct role in viral replication, as the RdRP is the main enzyme driving the replication process, whereas NSP3 and NSP5 are proteases that might have a more varied role and could therefore be less efficient to target [41].

During the SARS-CoV-2 pandemic, it became clear how much damage a viral outbreak can cause to our society. By designing flexible, broad-spectrum antivirals, we could be better prepared. CRISPR-Cas13b is sometimes called a reprogrammable approach, which means that with a few changes in the gRNA sequence we could target a new virus [5,6]. CRISPR-Cas13b is therefore a flexible system that could rapidly be adapted to a novel viral strain, something that is crucial when previously unknown viruses start to spread, as in the case of SARS-CoV-2 in late 2019. CRISPR-Cas13b has the potential to provide a rapid response to emerging viral diseases, which is of great need.

CRISPR-Cas has revolutionized the field of genetic engineering. CRISPR-based treatments of previously incurable genetic disorders are now obtaining approval for clinical use in humans [44,45]. Here, we showed that CRISPR-Cas13b is capable of targeting SARS-CoV-2 and SARS-CoV vRNAs with the same gRNAs, demonstrating that CRISPR-Cas could also act as a broad-spectrum antiviral.

## Figures and Tables

**Figure 1 viruses-16-01539-f001:**
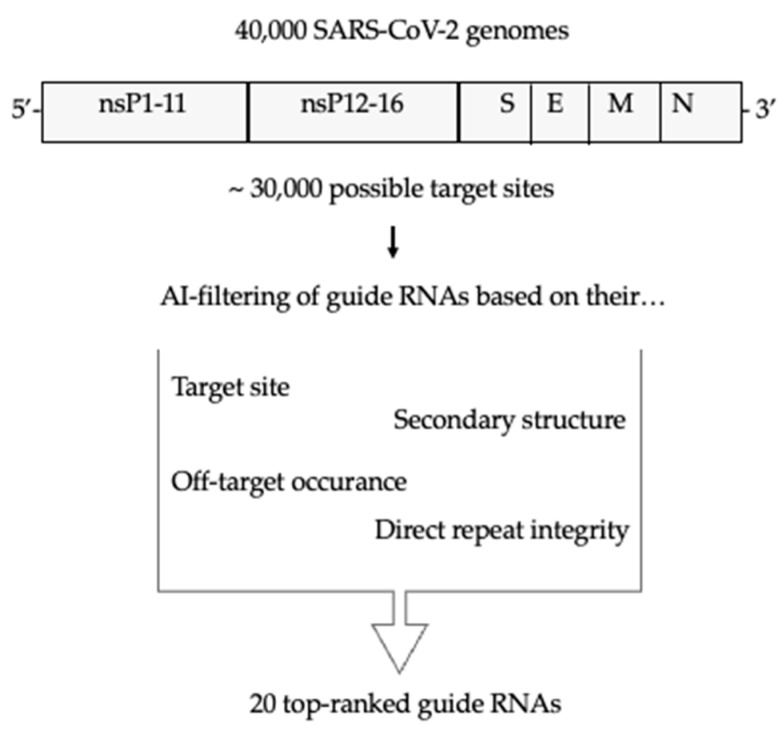
Overview of the selection and filtering of the most suited gRNAs to silence SARS-CoV-2. The gRNAs are filtered based on their target site conservation and relevance, secondary structure, host genome overlap, and direct repeat integrity. A total of 20 guide RNAs were chosen to be tested in vitro.

**Figure 2 viruses-16-01539-f002:**
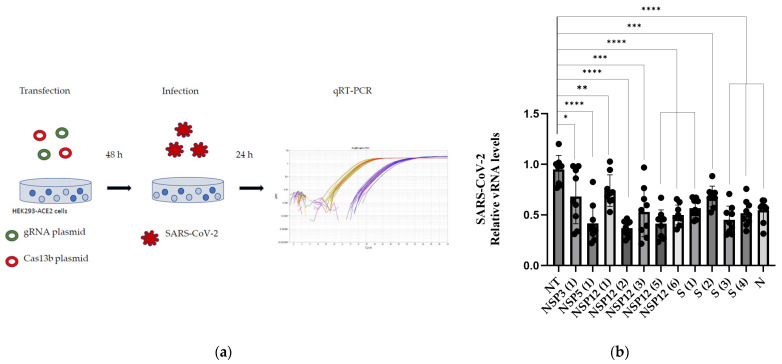
Single guide RNAs (gRNAs) silencing live SARS-CoV-2. (**a**) Scheme of the workflow for the screening of top-ranked gRNAs, including transfections of the plasmids encoding gRNA and Cas13b, infections with SARS-CoV-2, and analysis with qPCR. (**b**) The single gRNA silencing of SARS-CoV-2. n = 3, where n is the independent biological experiments. Statistical analysis performed with Student’s *t*-test. * *p* < 0.05, ** *p* < 0.01, *** *p* < 0.001, **** *p* < 0.0001.

**Figure 3 viruses-16-01539-f003:**
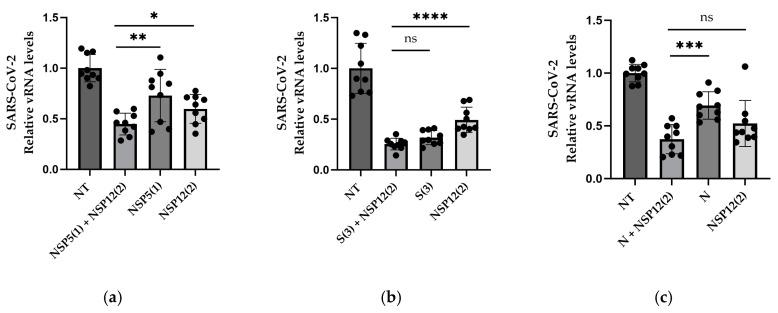
gRNAs tested in pairs; 50 ng of each gRNA plasmid tested together compared to 50 ng of the gRNAs tested individually. (**a**) NSP12-targeting gRNA together with NSP5 gRNA, (**b**) NSP12 gRNA together with S gRNA, and (**c**) NSP12 gRNA together with N gRNA. n = 3, where n is the independent biological experiments. Statistical analysis performed with Student’s *t*-test. * *p* < 0.05, ** *p* < 0.01, *** *p* < 0.001, **** *p* < 0.0001, ns. non-significant.

**Figure 4 viruses-16-01539-f004:**
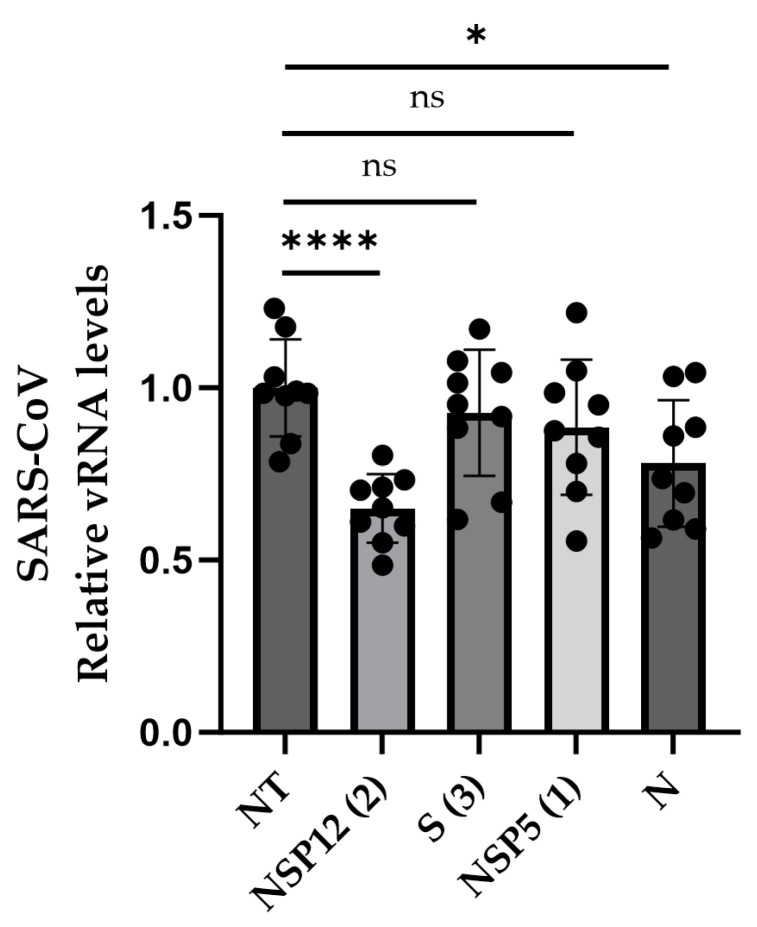
CRISPR-Cas13b designed to target SARS-CoV-2 efficacy on SARS-CoV. Statistical analysis performed with Student’s *t*-test. n = 3, where n is the independent biological experiments. * *p* < 0.05, **** *p* < 0.0001, ns: non-significant.

**Table 1 viruses-16-01539-t001:** The 20 selected top-ranked gRNAs, their target gene in the SARS-CoV-2 genome, and their sequences.

gRNA ID	Target	Sequence 5′ -> 3′
NSP3 (1)	NSP3	GACUAUCAUCAUCUAACCAAUCUUCUUCUU
NSP3 (2)	NSP3	CAACUUAGGGUCAAUUUCUGUACAAACAAC
NSP3 (3)	NSP3	GGAGGGUAAAAAGAACAAUACAUAUGUGAA
NSP3 (4)	NSP3	AUCCUUUCAUCAAGUUCAAAAGUGAUAUUC
NSP5 (1)	NSP5ab	GGUAAUUCCAUAUGGUGCAUGUAACAAAAA
NSP5 (2)	NSP5ab	GAACAUUACCAGCCUGUACCAAGAAAUUAU
NSP5 (3)	NSP5ab	AAAACGGCAAUUCCAGUUUGAGCAGAAAGA
NSP12 (1)	NSP12	GGCAUUAACAAUGAAUAAUAAGAAUCUACA
NSP12 (2)	NSP12	GAUGAAACUGUCUAUUGGUCAUAGUACUAC
NSP12 (3)	NSP12	GGUAAGGAAGGUACACAUAAUCAUCACCCU
NSP12 (4)	NSP12	GGGAUGACAUUACGUUUUGUAUAUGCGAAA
NSP12 (5)	NSP12	GCGAGCAAGAACAAGUGAGGCCAUAAUUCU
NSP12 (6)	NSP12	UAACGAUAGUAGUCAUAAUCGCUGAUAGCA
S (1)	Spike	GUCUGACUUCAUCACCUCUAAUUACAAAUG
S (2)	Spike	GGUGGUGUUUUGUAAAUUUGUUUGACUUGU
S (3)	Spike	AGGGUAAUAAACACCACGUGUGAAAGAAUU
S (4)	Spike	GGGUAAUAAACACCACGUGUGAAAGAAUUA
S (5)	Spike	GGUAAUUUAUAAUUAUAAUCAGCAAUCUUU
M	Membrane	GGAAAAUUAACUUAAUUAUAUACAAAAACC
N	Nucleocapsid	GGGAAUUUAAGGUCUUCCUUGCCAUGUUGA

**Table 2 viruses-16-01539-t002:** Ranked list of the guide RNA knockdown of SARS-CoV-2.

Guide RNA ID	Relative vRNA Levels	RNA Knockdown (%)
NSP12 (2)	0.37	63
NSP12 (5)	0.41	59
NSP5 (1)	0.41	59
S (3)	0.45	55
NSP12 (6)	0.5	50
S (4)	0.52	48
NSP12 (3)	0.53	47
N	0.54	46
S (1)	0.57	43
NSP3 (1)	0.68	32
S (2)	0.68	32
NSP12 (1)	0.74	26
NSP3 (2)	ns	ns
NSP3 (3)	ns	ns
NSP3 (4)	ns	ns
NSP5ab (2)	ns	ns
NSP5ab (3)	ns	ns
NSP12 (4)	ns	ns
S (5)	ns	ns

ns = non-significant results.

**Table 3 viruses-16-01539-t003:** Chosen gRNAs to test in further experiments, with their individual efficiency against SARS-CoV-2 and number of mismatches to SARS-CoV.

Guide RNA ID	RNA Knockdown (%)	Number of Mismatches to SARS-CoV
NSP12 (2)	63	1
NSP5 (1)	59	7
S (3)	55	12
N	46	4

**Table 4 viruses-16-01539-t004:** Summary of the SARS-CoV-2-targeting guide RNAs’ silencing of SARS-CoV and the correlation to the number of mismatches between the guide RNA and the target site in the SARS-CoV genome.

gRNA ID	Knockdown SARS-CoV-2 (%)	Number of Mismatches to SARS-CoV	Knockdown SARS-CoV (%)
NSP12 (2)	63	1	35
NSP5 (1)	59	7	ns
S (3)	55	12	ns
N	46	4	22

ns = non-significant results.

## Data Availability

The raw data supporting the conclusions of this article will be made available by the authors on request.

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
