# Peer review of "A CRISPR-Cas13b System Degrades SARS-CoV and SARS-CoV-2 RNA In Vitro"

_viruses, 2024, doi:10.3390/v16101539_

Round 1

Reviewer 1 Report

Comments and Suggestions for Authors

The submitted study examines a timely issue in the rapidly changing SARS-CoV-2 variant prevalence. The paper presents a novel use of an existing system and uses CRISPR-CAS13b system silencing. The AI algorithm designed for this purpose could greatly assist in the fight against future SARS-CoV-2 variants and other pandemic viruses by designing new antiviral therapies. The potential of using this technique to produce broad-spectrum antivirals already showed on SARS-CoV in the paper. Based on the data the hypothesis of the use of CRISPR-CAS13b silencing system as a potential antiviral method is well supported by the data. Although to use this method as a broad spectrum antiviral is only superficially touched upon in the Discussion. It is suggested to add some sentences for future use of the silencing system as broad-spectrum antiviral as tool for fighting against the pandemic.

The paper basically builds on the diversity of SARS-CoV-2 variants, but only one SARS-CoV-2 variant was tested with live virus. It is recommended that the wording of the manuscript be modified accordingly (introduction and discussion).

The Material and Methods section lacks a description of the source and variant distribution of the 40000 SARS-CoV-2 genomes used.  It is recommended that this be described in a separate section with identification of the public databases used for the study.

In Material and Methods section 2.2, it is suggested that the SARS-CoV-2 original or variant for Viral Strains be shown (lines 65-66)

In Material and Methods section 2.11, it is recommended to indicate the target gene for SARS-CoV-2 (line 143). At the description of SARS-CoV primers and probes SARS-CoV-2 need to correct SARS-CoV (line 148).

In the Results section is recommended to add relative vRNA levels to Table 2 to easier understand the connection to Figure 2b and show the significance level.

In the Results section In Figure 3. the brackets are missing from NSP12 and NSP5.

Reviewer 2 Report

Comments and Suggestions for Authors

1.        The research is original because it identifies the CRISPR-Cas13b system as an innovative tool in the fight against viral infections, including SARS-CoV, the virus responsible for COVID-19.

2.        This study adds to the understanding unlike other CRISPR systems that target DNA, Cas13b is specifically designed to cleave RNA. This makes it particularly effective against RNA viruses like SARS-CoV, crucial for addressing viral infections directly it represents a novel contribution.

3.        The author needs to explore using Cas13b in combination with other immune-modulating agents to enhance the host's antiviral response, providing a dual mechanism of action.

4.        Answer: Conclusions align with the presented results across different experiments.

5.        Are the references appropriate?

Answer: Yes

Recommendation: Published after minor revision.

Comments on the Quality of English Language

-

Reviewer 3 Report

Comments and Suggestions for Authors

The manuscript refers to a CRISPCast-13 system that degrades SARS coV-1 and SARS-CoV-2 in vitro. The rationale of the article is adequate, although it would be important for the reader to understand why non-structural proteins were part of the targets involved. In the method section, it is crucial to define how gRNA was generated based on the available data on SARS-CoV1 and 2 viruses. What was the efficiency of infection assays 2.10? Based on the results of Figure 2, Table 2 relates the gRNA with the highest efficiency of knockdown; strangely, there is only one data for gRNA; why? it should be a primary target. In Table 3, the number of mismatches may be expected, but not for NSP5; this point should be explained.  Please add the n values in each Figure. The results of NSP12 should be discussed further, and the results should be compared with those of NSP3 and NSP5. There are several limitations stated in the text; therefore, it would be valuable for the reader to have a section at the end in which the limitations are listed.
